# Passive Light Source Monitoring for Sending or Not Sending Twin-Field Quantum Key Distribution

**DOI:** 10.3390/e24050592

**Published:** 2022-04-23

**Authors:** Xuerui Qian, Chunhui Zhang, Huawei Yuan, Xingyu Zhou, Jian Li, Qin Wang

**Affiliations:** 1Institute of Quantum Information and Technology, Nanjing University of Posts and Telecommunications, Nanjing 210003, China; quentin_qxr@163.com (X.Q.); chz@njupt.edu.cn (C.Z.); jyyhw77@163.com (H.Y.); xyz@njupt.edu.cn (X.Z.); jianli@njupt.edu.cn (J.L.); 2Broadband Wireless Communication and Sensor Network Technology, Key Lab of Ministry of Education, NUPT, Nanjing 210003, China; 3Telecommunication and Networks, National Engineering Research Center, NUPT, Nanjing 210003, China

**Keywords:** twin-field quantum, sending-or-not-sending, passive light source monitoring

## Abstract

Twin-field quantum key distribution (TF-QKD) can break the repeaterless linear bound and possess the measurement-device-independent security, and thus seems very promising in practical applications of quantum secure communication. In most reported TF-QKD protocols, light sources are assumed to possess trusted and fixed photon number distributions (PND), which are unrealistic assumptions in practical applications. Fortunately, the light source monitoring (LSM) method is proposed to circumvent this problem by actively adjusting the attenuation coefficient and monitoring the photon number distribution of light sources. However, the active light source monitoring (ALSM) method may induce additional modulation errors due to imperfect attenuation devices, deteriorating practical performances of TF-QKD systems. In this manuscript, we propose a passive light source monitoring (PLSM) scheme for TF-QKD, and employ the sending-or-not-sending (SNS) TF-QKD as an example for illustration. Simulation results show that our present work can greatly exceed both the original SNS protocol and the ALSM scheme when light source fluctuations and modulation errors are taken into account.

## 1. Introduction

Quantum key distribution (QKD) allows two legitimate parties, Alice and Bob, to share secure keys based on the laws of quantum physics. The security of BB84 protocol was proven in theory by many scientists [1,2,3,4,5,6,7]. However, there are still some loopholes in the measurement devices. To remove those attacks directed on the measurement devices, the measurement-device-independent quantum key distribution (MDI-QKD) [8] was put forward. Thereafter, a lot of related experiments and theories have been done on MDI-QKD, making it more efficient and practical [9,10,11,12,13]—just its key rate is still limited by the fundamental limit of channel capacities without quantum repeaters, e.g., the Pirandola–Laurenza–Ottaviani–Banchi (PLOB) bound [14,15].

Recently, Lucamarini et al. proposed the twin-field quantum key distribution (TF-QKD) protocol [16], which can break the PLOB bound [14,15] and make the rate distance dependence change from a linear to square root. Up to date, many variants of TFQKD protocols have been proposed and experimentally demonstrated [17,18,19,20,21,22,23,24,25,26,27,28]. However, some assumptions are made for the sources in most reported works, i.e., with a trusted and fixed photon-number distribution (PND), which usually can not be satisfied in practical implementations. Those unreasonable assumptions will inevitably compromise the security of practical QKD systems. To solve the problem, the light source monitoring (LSM) method was put forward and experimentally realized by actively modulating local attenuator into different losses [29,30], hereafter called the active light source monitoring (ALSM) scheme. However, unfortunately, the ALSM scheme will bring new loopholes and then deteriorate practical performances of TF-QKD systems during intensity modulation processes due to imperfections of attenuated devices.

In this paper, we propose a passive light source monitoring (PLSM) scheme for TF-QKD, which is accomplished by a passive monitoring module consisting of a beam splitter and two detectors at the source side. Through the PLSM module, we can obtain four monitoring events by two local detectors and then precisely estimate the bounds of source distributions. Specifically, we employ the sending-or-not-sending (SNS) TFQKD [18,31,32,33] as an example for illustration. Compared with the ALSM method, our PLSM method can passively monitor the PND and dramatically exceed the performance of ALSM when modulation errors are considered.

## 2. PLSM Scheme in SNS−TFQKD

In this section, we describe the SNS−TFQKD scheme [18,31,32,33] with PLSM. The schematic of the setup is shown in Figure 1, where it involves two senders, Alice and Bob, and one untrustworthy third party (UTP), Charlie. The detailed process of the SNS−TFQKD with a four-intensity decoy-state PLSM scheme can be described as follows:

*Step 1*. Alice and Bob do some preparatory work before sending the pulse: they send a reference coherent light to Charlie, and Charlie performs phase compensation.

*Step 2*. The *N* pulses generated by Alice and Bob are encoded by a modulation module, which contains a phase modulator (PM) and an intensity modulator (IM). During this process, each pulse is randomly chosen as the *X* (decoy) or *Z* (signal) window. At any time window *i*, Alice (Bob) independently determines whether it is a decoy window or a signal window. If the decoy window is chosen, she (he) prepares state $|\sqrt{v_{k}}e^{i\delta_{Ai}+i\gamma_{Ai}}\rangle \left(|\sqrt{v_{k}}e^{i\delta_{Bi}+i\gamma_{Bi}}\rangle \right)$ and sends it to Charlie, $v_{k}$
(k=0,1,2). If the signal window is selected, Alice (Bob) sends states $|\sqrt{u}e^{i\delta_{Ai}+i\gamma_{Ai}}\rangle \left(|\sqrt{u}e^{i\delta_{Bi}+i\gamma_{Bi}}\rangle \right)$ to Charlie with a probability of ε, and with a probability of (1−ε) for not sending, where the $\gamma_{A},\gamma_{B}$ are global phases of the coherent states.

*Step 3*. The pulses are split into two modes, where the idler mode is for performing PLSM and the signal mode is for encoding information and sending it to Charlie. The idler mode is further split by a local beam splitter and then sent into two local detectors. As a result, four detection events can be obtained. For example, in Alice’s PLSM module, these events can be denoted as *l*(l=x,y,z,w): *x*, neither $D_{1}$ or $D_{2}$ clicks; *y*, only $D_{1}$ clicks; *z*, only $D_{2}$ clicks; *w*, and both $D_{1}$ and $D_{2}$ click.

*Step 4*. Charlie measures all received states with a beam splitter and two detectors ($D_{L}$ and $D_{R}$), and then announces the effective measurement outcome, i.e., which detector clicks.

*Step 5*. Alice and Bob announce the local detection events *l* and the kind of window (*X* window or *Z* window) for each pulse. In addition, the intensity and encoding phases ($\delta_{A}$, $\delta_{B}$) in the *X* basis should also be disclosed.

*Step 6*. Alice and Bob use the data in the *X* basis to estimate channel parameters, and they randomly select some bits in the *Z* basis for the error test and use the remaining bits to extract the final secure keys.

In this protocol, *Z* basis is defined as the time window when both Alice and Bob choose the signal window; X basis is denoted as the time window when both Alice and Bob choose the same decoy window (with intensity $v_{k}$), and, simultaneously, the random phases $\delta_{A}$, $\delta_{B}$ prepared in the window satisfy
(1)$$1-\left|\cos\left(\delta_{A}-\delta_{B}\right)\right|\leq \left|\lambda \right|.$$Here, λ is determined by the size of the phase slice chosen by Alice and Bob. In addition, the effective measurement outcome denotes that only one detector ($D_{L}$ or $D_{R}$) clicks.

In the PLSM module, when the event *l* occurs, the idler state is projected into $\rho =\sum P_{n}\left(\mu \right)q_{n}^{l}\left|n\rangle \langle n\right|$, where $P_{n}\left(\mu \right)$ denotes the photon number distribution of the weak coherent state (WCS) with mean photon number μ
$\left(\mu \in \left\{v_{0},v_{1},v_{2},u\right\}\right)$, $P_{n}\left(\mu \right)=e^{-\mu }\frac{\mu^{n}}{n!}$, and $q_{n}^{l}$ is the probability of an *n*-photon state projecting into event *l* given by [34,35]
(2)$$\begin{array}{cc} \; & q_{n}^{x}={\left(1-d_{s}\right)}^{2}{\left(1-\eta_{s}\right)}^{n},\\ \; & q_{n}^{y}=\left(1-d_{s}\right){\left(1-\eta_{s}\right)}^{n}\left[{\left(1+\frac{t\eta_{s}}{1-\eta_{s}}\right)}^{n}+d_{s}-1\right],\\ \; & q_{n}^{z}=\left(1-d_{s}\right){\left(1-\eta_{s}\right)}^{n}\left[{\left(\frac{1-t\eta_{s}}{1-\eta_{s}}\right)}^{n}+d_{s}-1\right],\\ \; & q_{n}^{w}=1-q_{n}^{x}-q_{n}^{y}-q_{n}^{z}, \end{array}$$
where $d_{s}$ and $\eta_{s}$ are the dark counting rate and detection efficiency of the local detectors at the sender’s side (Alice and Bob), respectively, and *t* is the transmittance of ${\mathrm{BS}}_{2}$ and ${\mathrm{BS}}_{4}$. Here, for simplicity, we assume that two local detectors in PLSM module have the same detection efficiency and dark count rate, which means $\eta_{1}=\eta_{2}=\eta_{s}$, $d_{1}=d_{2}=d_{s}$. Define $a_{n}^{l}\left(\mu \right):=P_{n}\left(\mu \right)q_{n}^{l}$ as the photon number distribution under different counting events.

Then, we estimate the upper and lower bounds of probabilities of different photon-number states, i.e., the vacuum state, the one-photon state and the two-photon state. By measuring the idler mode, the gain of four events can be obtained as
(3)$$\begin{array}{cc} \; & Q_{x}\left(\mu \right)=\sum a_{n}^{x}\left(\mu \right)=\sum e^{-\mu }\frac{\mu^{n}}{n!}{\left(1-d_{s}\right)}^{2}{\left(1-\eta_{s}\right)}^{n},\\ \; & Q_{y}\left(\mu \right)=\sum a_{n}^{y}\left(\mu \right)=\sum e^{-\mu }\frac{\mu^{n}}{n!}\left(1-d_{s}\right){\left(1-\eta_{s}\right)}^{n}\left[{\left(1+\frac{t\eta_{s}}{1-\eta_{s}}\right)}^{n}+d_{s}-1\right],\\ \; & Q_{z}\left(\mu \right)=\sum a_{n}^{z}\left(\mu \right)=\sum e^{-\mu }\frac{\mu^{n}}{n!}\left(1-d_{s}\right){\left(1-\eta_{s}\right)}^{n}\left[{\left(\frac{1-t\eta_{s}}{1-\eta_{s}}\right)}^{n}+d_{s}-1\right],\\ \; & Q_{w}\left(\mu \right)=\sum a_{n}^{w}\left(\mu \right)=1-Q_{x}\left(\mu \right)-Q_{y}\left(\mu \right)-Q_{z}\left(\mu \right). \end{array}$$According to the derivation presented in Appendix A, the estimations of $P_{n}^{L}\left(\mu \right)$, $P_{n}^{U}\left(\mu \right)$
(n=0,1,2) are given by
(4)$$\begin{array}{c} P_{0}^{L}\left(\mu \right)=P_{0}^{U}\left(\mu \right)=\frac{Q_{x}\left(\mu \right)}{{\left(1-d_{s}\right)}^{2}}, \end{array}$$
(5)$$\begin{array}{c} P_{1}^{L}\left(\mu \right)=\frac{q_{2}^{z}Q_{y}\left(\mu \right)-q_{2}^{y}Q_{z}\left(\mu \right)-\left(q_{2}^{z}q_{0}^{y}-q_{0}^{z}q_{2}^{y}\right)P_{0}^{L}\left(\mu \right)}{q_{2}^{z}q_{1}^{y}-q_{1}^{z}q_{2}^{y}}, \end{array}$$
(6)$$\begin{array}{c} P_{1}^{U}\left(\mu \right)=\frac{\left(q_{2}^{y}-q_{3}^{y}\right)Q_{z}\left(\mu \right)-q_{2}^{z}Q_{y}\left(\mu \right)+\left(q_{2}^{z}q_{0}^{y}-q_{2}^{z}q_{3}^{y}-q_{0}^{z}q_{2}^{y}+q_{0}^{z}q_{3}^{y}\right)P_{0}^{L}\left(\mu \right)+q_{2}^{z}q_{3}^{y}}{q_{1}^{y}\left(q_{2}^{y}-q_{3}^{y}\right)-q_{2}^{y}\left(q_{1}^{y}-q_{3}^{y}\right)}, \end{array}$$
(7)$$\begin{array}{c} P_{2}^{L}\left(\mu \right)=\frac{Q_{y}\left(\mu \right)-\left(q_{0}^{y}-q_{3}^{y}\right)P_{0}^{U}\left(\mu \right)-\left(q_{1}^{y}-q_{3}^{y}\right)P_{1}^{U}\left(\mu \right)-q_{3}^{y}}{q_{2}^{y}-q_{3}^{y}}, \end{array}$$
(8)$$\begin{array}{c} P_{2}^{U}\left(\mu \right)=\frac{Q_{z}\left(\mu \right)-q_{0}^{z}P_{0}^{L}\left(\mu \right)-q_{1}^{z}P_{1}^{L}\left(\mu \right)}{q_{2}^{z}}, \end{array}$$
where $P_{n}^{L(U)}\left(\mu \right)$ represents the lower or upper bounds on the probability of having the *n*-photon state given the mean photon number μ.

In the SNS−TFQKD, Alice and Bob simultaneously send photon pulses to the untrustworthy third party (UTP) Charlie. According to Ref. [36], the decoy-state method is still applicable under unknown PND conditions; the lower bound of the single-photon counting rate and the upper bound of the single-photon error rate can be estimated as
(9)$$s_{1}^{L}=\frac{p_{2}^{L}\left(v_{2}\right)\left[S_{v_{1}}-p_{0}^{U}\left(v_{1}\right)S_{0}\right]-p_{2}^{U}\left(v_{1}\right)\left[S_{v_{2}}-p_{0}^{L}\left(v_{2}\right)S_{0}\right]}{p_{2}^{U}\left(v_{2}\right)p_{1}^{U}\left(v_{1}\right)-p_{2}^{L}\left(v_{1}\right)p_{1}^{L}\left(v_{2}\right)},$$
(10)$$e_{1}^{ph,U}=\frac{S_{v_{1}}E_{v_{1}}-p_{0}^{L}\left(v_{1}\right)S_{0}/2}{p_{1}^{L}\left(v_{1}\right)s_{1}^{L}}.$$Here, we set $v_{0}=0$, $v_{2}>v_{1}>0$, and $S_{v_{k}}$, $E_{v_{k}}$ are the counting rate and the bit error rate of a state with intensity $v_{k}$ sent in decoy windows, respectively. In addition, the relationships between $P_{n}^{L(U)}\left(\mu \right)$ and $p_{n}^{L(U)}\left(\mu \right)$ are set by [30]
(11)$$\begin{array}{cc} \; & p_{0}^{L(U)}\left(\mu \right):={\left[P_{0}^{L(U)}\left(\mu \right)\right]}^{2},p_{1}^{L(U)}\left(\mu \right):=2P_{0}^{L(U)}\left(\mu \right)P_{1}^{L(U)}\left(\mu \right),\\ \; & p_{2}^{L(U)}\left(\mu \right):=2P_{0}^{L(U)}\left(\mu \right)P_{2}^{L(U)}\left(\mu \right)+{\left[P_{1}^{L(U)}\left(\mu \right)\right]}^{2}. \end{array}$$Finally, the secure key rate is
(12)$$R=2\varepsilon \left(1-\varepsilon \right)P_{1}^{L}\left(u\right)s_{1}^{L}\left(1-H\left(e_{1}^{ph,U}\right)\right)-S_{Z}fH\left(E_{Z}\right),$$
where ε represents the probability that Alice (Bob) chooses to send out a signal pulse (it can be preset in the protocol); $H\left(x\right)=-x\log_{2}x-\left(1-x\right)\log_{2}\left(1-x\right)$ is the binary Shannon entropy function; $P_{1}^{L}\left(u\right)$ is the lower bound of the probability of single photons in the signal state; $S_{Z}$ and $E_{Z}$ refer to the gain and the average quantum bit error of pulses with intensity *u* sent in signal windows.

## 3. Numerical Simulations and Analysis

In the following, we perform numerical simulations for the original SNS−TFQKD [18], the SNS−TFQKD with ALSM [30] and the SNS−TFQKD with PLSM. In simulations, the gain and the quantum bit error of decoy states in *X* basis in Equations (9) and (10) are expressed as
(13)$$\begin{array}{cc} \; & S_{v_{k}}=S_{v_{k}}^{C}+S_{v_{k}}^{E},\\ \; & S_{v_{k}}E_{v_{k}}=E_{opt}S_{v_{k}}^{C}+\left(1-E_{opt}\right)S_{v_{k}}^{E},\\ \; & S_{v_{k}}^{C}=\frac{1}{\Delta }\left(1-P_{dc}\right)\int_{-\Delta /2}^{\Delta /2}e^{-v_{k}\eta \left(1-\cos\delta \right)}d\delta -{\left(1-P_{dc}\right)}^{2}e^{-2v_{k}\eta },\\ \; & S_{v_{k}}^{E}=\frac{1}{\Delta }\left(1-P_{dc}\right)\int_{-\Delta /2}^{\Delta /2}e^{-v_{k}\eta \left(1+\cos\delta \right)}d\delta -{\left(1-P_{dc}\right)}^{2}e^{-2v_{k}\eta }, \end{array}$$
where $S_{v_{k}}^{C}\left(S_{v_{k}}^{E}\right)$ is the counting rate of twin-field states entering into the correct (wrong) detector; $\eta =\eta_{D}10^{-\frac{\alpha s}{20}}$ denotes the total channel transmittance, where α and *s* are the loss coefficient and the length of channels, respectively; $\eta_{D}$ and $P_{dc}$ each refer to the efficiency and the dark count rate of detectors at Charlie’s side, respectively. $\delta =\left|\delta_{B}-\delta_{A}\right|$ is the phase difference between the twin-field states prepared by Alice and Bob; Δ=2π/M is the size of the phase slice and M is the number of phase slices; $E_{opt}$ represents the optical misalignment error, $S_{0}$ is the counting rate of the vacuum state, and $S_{0}=2P_{dc}\left(1-P_{dc}\right)$. The average quantum bit error and the counting rate for the signal states can be respectively expressed as:(14)$$\begin{array}{cc} \; & S_{Z}E_{Z}=2P_{dc}\left(1-P_{dc}\right){\left(1-\varepsilon \right)}^{2}+2\varepsilon^{2}\left(1-P_{dc}\right)e^{-u\eta }\left[I_{0}\left(u\eta \right)-\left(1-P_{dc}\right)e^{-u\eta }\right],\\ \; & S_{Z}=4\varepsilon \left(1-\varepsilon \right)\left(1-P_{dc}\right)e^{-\frac{u\eta }{2}}\left[1-\left(1-P_{dc}\right)e^{-\frac{u\eta }{2}}\right]+S_{Z}E_{Z}. \end{array}$$Here, $I_{0}\left(x\right)$ is the 0-order hyperbolic Bessel function of the first kind [31].

In the following, we do comparisons among three schemes, the original SNS−TFQKD, the ALSM SNS−TFQKD, and the present PLSM SNS−TFQKD, by using either ideal light sources without intensity fluctuations or practical light sources with intensity fluctuations. In addition, we also analyze the effect of modulation error on the ALSM scheme. The basic device parameters are shown in Table 1 [30].

In ALSM, the attenuation coefficients are set as $\eta_{0}=1$, $\eta_{1}=0.95$, $\eta_{2}=0.9$. For a fair comparison, in our scheme, the detection efficiency in PLSM module is set as $\eta_{s}=0.9$. In addition, the dark count rate of local detectors $d_{s}$ in ALSM and PLSM is set with the same value denoted as $P_{dc}$ as listed in row *a* of Table 1. In addition, we adopt the local search algorithm (LSA) [9] to optimize the parameters $\epsilon ,t,v_{0},v_{1},v_{2},u$. The numerical simulation results are presented in Figure 2, Figure 3 and Figure 4.

The performance of different monitoring methods with ideal sources without intensity fluctuations are presented in Figure 2. Simulation results show that the performance of our proposed PLSM SNS−TFQKD is comparable to both the original SNS−TFQKD and the ALSM SNS−TFQKD. It is also clear that the maximum transmission distance of all schemes exceeds 800 km. In addition, if actively odd-parity pairing (AOPP) [33] of post data processing is adopted, it can further improve the distance and key rate of SNS−TFQKD significantly.

However, in realistic implementations, the fluctuation of light sources is a common phenomenon in QKD systems [37,38]; therefore, it should be taken into account. In general, the signal from light sources can be considered as a coherent state, whose intensity usually possesses a Gaussian distribution:(15)$$G\left(\mu \right)=\frac{1}{\sqrt{2\pi }\sigma_{\mu }}\exp\left[-\frac{{\left(\mu -\mu_{0}\right)}^{2}}{2\sigma_{\mu }^{2}}\right],$$
where $\mu_{0}$ and $\sigma_{\mu }$ represent the mean value and standard deviation, respectively. Define the fluctuation coefficient as $\sigma :=\sigma_{\mu }/\mu_{0}$, and $Q_{l}\left(\mu \right)$ can be rewritten as
(16)$$Q_{l}\left(\mu_{0}\right)=\int Q_{l}\left(\mu \right)G\left(\mu \right)d\mu .$$

After calculating and simplifying, the counting rate of four events can be reformulated as
(17)$$\begin{array}{c} Q_{x}\left(\mu_{0}\right)={\left(1-d_{s}\right)}^{2}\exp\left[\frac{1}{2}\eta_{s}\left(-2\mu_{0}+\eta_{s}\sigma_{\mu }^{2}\right)\right],\\ Q_{y}\left(\mu_{0}\right)=\left(1-d_{s}\right)\exp\left(-\frac{\mu_{0}^{2}}{2\sigma_{\mu }^{2}}\right)\left\{\left(1-d_{s}\right)\exp\left[\frac{{\left(\mu_{0}-\eta_{s}\sigma_{\mu }^{2}\right)}^{2}}{2\sigma_{\mu }^{2}}\right]-\exp\left[\frac{{\left(\mu_{0}-\left(1-t\right)\eta_{s}\sigma_{\mu }^{2}\right)}^{2}}{2\sigma_{\mu }^{2}}\right]\right\},\\ Q_{z}\left(\mu_{0}\right)=\left(1-d_{s}\right)\exp\left(-\frac{\mu_{0}^{2}}{2\sigma_{\mu }^{2}}\right)\left\{\left(1-d_{s}\right)\exp\left[\frac{{\left(\mu_{0}-\eta_{s}\sigma_{\mu }^{2}\right)}^{2}}{2\sigma_{\mu }{}^{2}}\right]-\exp\left[\frac{{\left(\mu_{0}-t\eta_{s}\sigma_{\mu }{}^{2}\right)}^{2}}{2\sigma_{\mu }{}^{2}}\right]\right\},\\ Q_{w}\left(\mu_{0}\right)=1-Q_{x}\left(\mu_{0}\right)-Q_{y}\left(\mu_{0}\right)-Q_{z}\left(\mu_{0}\right). \end{array}$$

On the other hand, $S_{v_{k}}^{C}$ and $S_{v_{k}}^{E}$ will change if light fluctuations are considered, which can be expressed as [36]
(18)$$\begin{array}{c} {\hat{S}}_{v_{k}}^{C}=\int S_{v_{k}}^{C}G\left(v_{k}\right)dv_{k}=\int \left[\frac{1}{\Delta }\left(1-P_{dc}\right)\int_{-\Delta /2}^{\Delta /2}e^{-v_{k}\eta \left(1-\cos\delta \right)}d\delta -{\left(1-P_{dc}\right)}^{2}e^{-2v_{k}\eta }\right]G\left(v_{k}\right)dv_{k},\\ {\hat{S}}_{v_{k}}^{E}=\int S_{v_{k}}^{E}G\left(v_{k}\right)dv_{k}=\int \left[\frac{1}{\Delta }\left(1-P_{dc}\right)\int_{-\Delta /2}^{\Delta /2}e^{-v_{k}\eta \left(1+\cos\delta \right)}d\delta -{\left(1-P_{dc}\right)}^{2}e^{-2v_{k}\eta }\right]G\left(v_{k}\right)dv_{k}. \end{array}$$Then, the counting rate and the quantum bit error of decoy states in *X* basis can be respectively re-expressed as
(19)$$\begin{array}{c} {\hat{S}}_{v_{k}}={\hat{S}}_{v_{k}}^{C}+{\hat{S}}_{v_{k}}^{E},\\ {\hat{S}}_{v_{k}}{\hat{E}}_{v_{k}}=E_{opt}{\hat{S}}_{v_{k}}^{C}+\left(1-E_{opt}\right){\hat{S}}_{v_{k}}^{E}. \end{array}$$Then, $s_{1}^{L}$ and $e_{1}^{ph,U}$ can be rewritten as [7]
(20)$$s_{1}^{L}=\frac{p_{2}^{L}\left(v_{2}\right)\left[{\hat{S}}_{v_{1}}-p_{0}^{U}\left(v_{1}\right)S_{0}\right]-p_{2}^{U}\left(v_{1}\right)\left[{\hat{S}}_{v_{2}}-p_{0}^{L}\left(v_{2}\right)S_{0}\right]}{p_{2}^{U}\left(v_{2}\right)p_{1}^{U}\left(v_{1}\right)-p_{2}^{L}\left(v_{1}\right)p_{1}^{L}\left(v_{2}\right)},$$
(21)$$e_{1}^{ph,U}=\frac{{\hat{S}}_{v_{1}}{\hat{E}}_{v_{1}}-p_{0}^{L}\left(v_{1}\right)S_{0}/2}{p_{1}^{L}\left(v_{1}\right)s_{1}^{L}}.$$

In the following, we compare the performance of SNS−TFQKD with various schemes under different fluctuation coefficients σ.

In order to simulate the realistic condition, we use a set of practical system parameters in row *b* of Table 1 [16]. In the realistic condition, the coefficient of light intensity fluctuation σ is usually greater than 1% [39]; therefore, we set coefficient of intensity fluctuations as σ=1% and σ=2%.

As we can see from Figure 3a,b, when intensity fluctuations are taken into account, both ALSM and our present PLSM can show much better performance compared with the original SNS TF-QKD protocol. For example, when σ=1% or 2%, both ALSM and our present PLSM only slightly decrease its key rate and transmission distance, while the original SNS TF-QKD protocol rapidly drops its key rate and transmission distance.

In practice, when switching between different coefficients, it may bring into modulation errors in the ALSM scheme. We define the attenuation coefficients modulation error as Ω, then $\eta^{U}=\eta \left(1+\Omega \right)$, $\eta^{L}=\eta \left(1-\Omega \right)$, where $\eta^{U}$ and $\eta^{L}$ are the upper bound and lower bound of attenuation coefficients. Based on [30], $P_{n}^{L(U)}\left(\mu \right)$ in ALSM has been reestimated in Appendix B. The simulation results are shown in Figure 4, and it is obvious that our scheme can show much better performance than the ALSM scheme when the modulation error is accounted. For example, when we reasonably set Ω=0.02%, the maximum transmission distance drops from 540 km to 435 km for the ALSM scheme, while it still remains constant for our PLSM scheme. Obviously, the ALSM scheme is very susceptible to intensity modulation errors.

## 4. Conclusions

In this paper, we propose a PLSM scheme for TF-QKD, which is accomplished by implementing a passive monitoring module consisting of a beam splitter and two detectors at the source side. Through the PLSM module, we can obtain four kinds of monitoring events with two local detectors and can then precisely estimate the bounds of source distributions. We build a theoretical model and carry out corresponding numerical simulations. Simulation results show that our present work can outperform the original SNS−TFQKD protocol when there are existing intensity fluctuations in the light sources. Moreover, it shows much better performance than the reported ALSM scheme when modulation errors are taken into account. Therefore, our present work can not only reduce assumptions on the source distribution in former TF-QKD protocols, but also close the additional loopholes existing in the former active monitoring scheme, and thus seems very promising in practical implementations of QKD in the near future. 

## Figures and Tables

**Figure 1 entropy-24-00592-f001:**
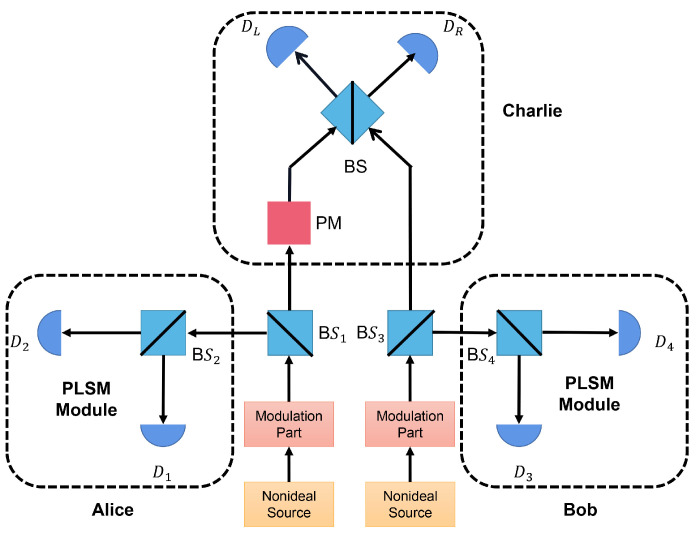
Schematic of the SNS−TFQKD system with PLSM. The PLSM module consists of a beam splitter (BS) and two local single-photon detectors.

**Figure 2 entropy-24-00592-f002:**
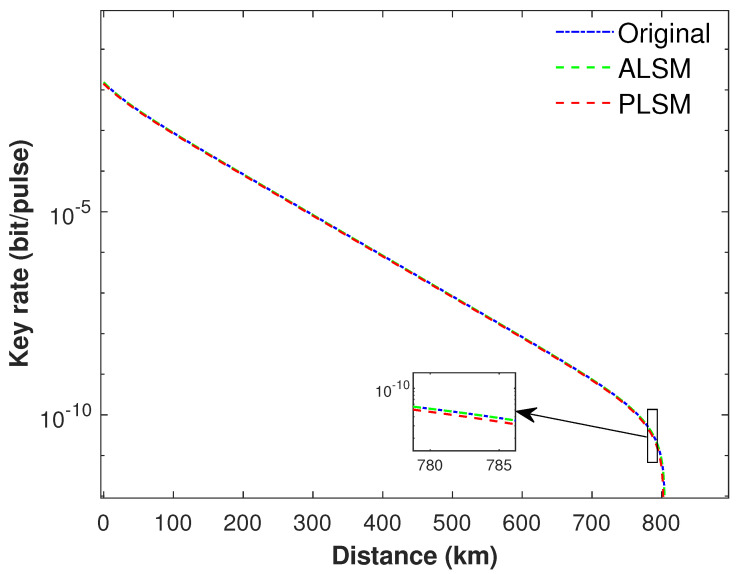
The secure key rate of different monitoring method using a set of parameters in row *a* of Table 1. The blue solid curve represents the original SNS−TFQKD; the red dash curve denotes our present PLSM SNS−TFQKD; and the green dash curve is the ALSM SNS−TFQKD. In addition, the variation trend of the key rate ranging between 780 km and 785 km is illustrated in the inset. It shows that our proposal can offer a key rate comparable to both the original SNS−TFQKD and the ALSM SNS−TFQKD when using photon sources without fluctuations.

**Figure 3 entropy-24-00592-f003:**
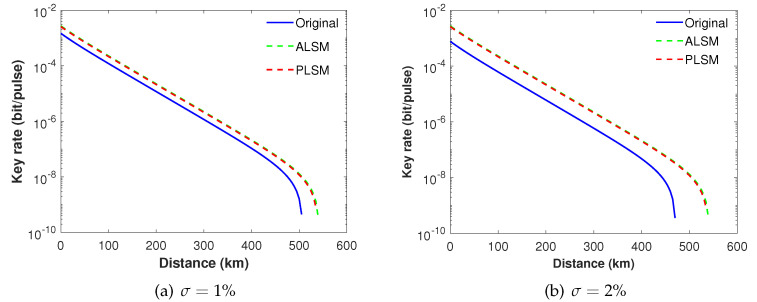
The secure key rate of different schemes with intensity fluctuation σ=1% in (**a**) and σ=2% in (**b**) when intensity modulation errors are not taken into account.

**Figure 4 entropy-24-00592-f004:**
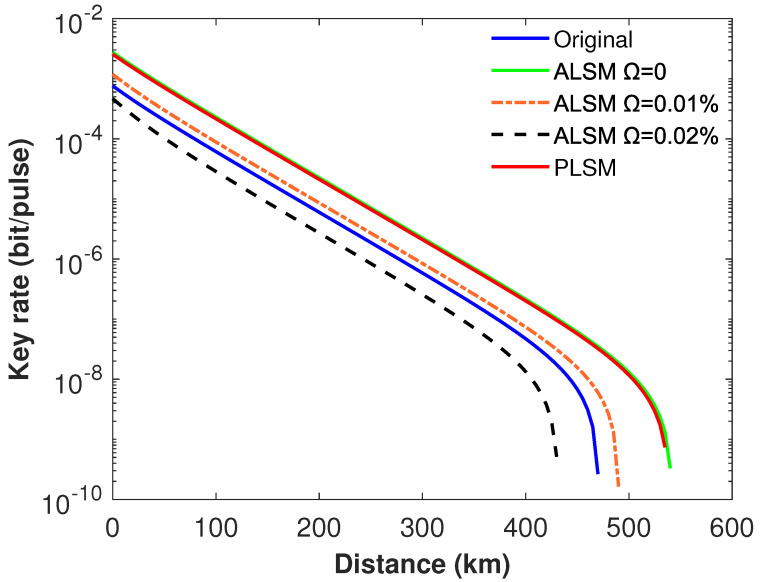
The secure key rate of different monitoring methods when different modulation errors are taken into account. Here, parameters in line *b* of Table 1 and the intensity fluctuation δ=2% are used. The blue solid curve represents the original SNS−TFQKD; the green solid curve is the ALSM SNS−TFQKD when Ω=0; the red solid curve denotes the PLSM SNS−TFQKD; the orange dotted-dash curve and the black dash curve denote the ALSM SNS−TFQKD with modulation errors Ω=0.01% and Ω=0.02%, respectively.

**Table 1 entropy-24-00592-t001:** The basic system parameters used in our numerical simulations. α: the loss coefficient of fiber at telecommunication wavelength (dB/km); $\eta_{D}$ and $P_{dc}$ are the efficiency and dark count rate of detectors at Charlie’s side; $E_{opt}$: the misalignment error of the QKD system; *f*: the error correction efficiency; *M*: the number of phase slices.

	α	*M*	$\eta_{D}$	$P_{\mathrm{dc}}$	$E_{\mathrm{opt}}$	*f*
set *a*	0.2 dB/km	16	80%	$10^{-11}$	1%	1.1
set *b*	0.2 dB/km	16	30%	$10^{-9}$	3%	1.15

## Data Availability

Not applicable.

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
