# Peer review of "Passive Light Source Monitoring for Sending or Not Sending Twin-Field Quantum Key Distribution"

_entropy, 2022, doi:10.3390/e24050592_

Round 1
Reviewer 1 Report
The authors propose a passive light source monitoring scheme for twin-field quantum key distribution. They built a theoretical model and carried out numerical simulations with aim to employ the sending-or-not-sending protocol when there existing intensity fluctuations in the light sources, showing better performance than the reported active light source monitoring. I think the results of this manuscript can be published after the authors give clearer description of the method used in the numerical simulations performed.
Reviewer 2 Report
In this article the authors have studied "Passive light source monitoring for sending-or-not-sending twin-field quantum key distribution". My comments regarding the publication of this paper are appended below point-wise:
- The paper is well written and the derived results and the findings are clearly presented.
- Derived results are correct.
- Reference list is complete.
- Numerical plots are clearly presented.
Based on the above mentioned points I would strongly recommend publication of this paper.
